# Dental Students’ Experience, Impact, and Response to Patient Aggression in Saudi Arabia: A Nationwide Study

**DOI:** 10.3390/healthcare10112239

**Published:** 2022-11-09

**Authors:** Sanjeev B. Khanagar, Ibrahim Aldawas, Abdullah Almutairi, Mohammed Alamro, Naif Altammami, Sultan Aldakhil, Suliman Hodan

**Affiliations:** 1Preventive Dental Science Department, College of Dentistry, King Saud Bin Abdulaziz University for Health Sciences, Riyadh 11426, Saudi Arabia; 2King Abdullah International Medical Research Centre, Ministry of National Guard Health Affairs, Riyadh 11481, Saudi Arabia; 3College of Public Health, Texila American University, Georgetown 413741, Guyana; 4College of Dentistry, King Saud Bin Abdulaziz University for Health Sciences, Riyadh 11426, Saudi Arabia

**Keywords:** patient aggression, violence, dental students, dental professionals, impact, response, workplace violence, assault, abuse, mistreatment

## Abstract

Patient aggression and violence comprise a wide range of behaviors and actions that may include verbal aggression and physical aggression. The aim of this study is to report dental students’ experience with, impact from, and response to patient aggression in Saudi Arabia. A cross-sectional analytical study was conducted among dental students from various health universities representing each region of Saudi Arabia. Data were collected using a self-administered, structured and validated questionnaire. A total of 375 participants responded to the questionnaire and 121 (32.3%) study participants reported experiencing patient aggression. Out of those, 91 (75.21%) experienced patients displaying anger or raising their voice toward them, 37 (30.58%) reported being insulted by a patient, 22 (18.18%) reported being threatened, 12 (6.2%) had experienced sexual harassment, and 65 (50.41%) had experienced verbal harassment. Furthermore, 91 (75.21%) participants reported being abandoned by patients because they were students. A total of 55 (45.45%) participants stated that aggressive patient behavior had an impact on their clinical performance, and 44 (36.36%) considered themselves stressed out. A total of 24 (19.83%) participants took time off due to incidents and 22 (18.18%) thought about quitting dentistry. Prevalence of patient aggression was significant among dental students in Saudi Arabia. These episodes of patient aggression negatively impacted students’ academic performance and wellbeing, necessitating urgent attention. Educational institutions should conduct periodic workshops for students in order to address these issues. Policymakers should develop better policies in order to reduce violence and aggression against health care providers.

## 1. Introduction

Violence against health care providers is very common and considered a global issue. Surprisingly, 8 to 38% of health care providers experience some form of violence at some point of time in their careers. Most times, they are either threatened or exposed to verbal or physical aggression [1]. Workplace violence mainly includes violent acts, which present in the form of physical assaults or threats of assaults directed towards a person at work or on duty [2]. Patient aggression and violence comprise a wide range of behaviors and actions that may include verbal aggression (yelling or swearing), intimidating and threatening, abusive letters, phone calls or messages, online trolling, inappropriate body language, assault, indecent behavior, sexual assault, stalking, and many more [3]. Health care providers encounter these aggressive behaviors, which negatively affect wellbeing, either from their patients or from the patients’ relatives [4,5,6,7]. The World Medical Association considers violence against health care providers an international emergency that undermines the very foundation of health systems and critically impacts patient health [8].

Several studies have reported experience of patient aggression among health care providers; however, the majority of these studies have been conducted among nurses and other health care providers [9,10,11,12]. A very limited number of studies have reported experiences of patient aggression among dental professionals. A study conducted among Nigerian dental professionals reported that 31.9% of study participants had experienced violence from patients or their relatives [13]. Another study reported experience of challenging patients among German dentists; 25% of study participants reported experiencing aggressive and demanding patients [14]. These studies highlighted long waiting time, cancellation of appointments, and unsatisfactory treatment outcomes as a few of the reasons for patients’ aggressive behavior [13,14]. Another study reported occupational health hazards among dental practitioners in New Zealand; 26% of study participants who were solo practitioners had experienced violent and abusive incidents [15]. Another study reported the British Dental Association survey; 80% of participants, including both dentists and dental auxiliaries, had experienced some type of aggression at work [16].

Still another study reported experience of patient aggression among dental students in the United States; the authors reported that 28% of study participants had experienced at least one instance of physical aggression, and 86% of them had experienced at least one instance of verbal aggression [17]. Another study reported experience of sexual harassment from patients among Brazilian undergraduate dental-school students; 9.1% of study participants reported experiencing sexual harassment from patients [18]. A recent study reported prevalence and impact of harassment against Australasian dental students from their patients; 20% of study participants experienced at least one incident of patient aggression [19].

Experiences of patient aggression and harassment may have severe consequences on health care professionals. These may include post-traumatic stress disorder (PTSD), taking time off work, demotivation, decreased job satisfaction, lower self-confidence, increased anxiety levels, dissatisfaction, and sleep disturbances, all of which can affect professionals’ performance [20,21,22].

Studies that have reported patient aggression toward health-professional students indicated that these experiences can negatively impact their confidence, performance, and overall career. Moreover, students may not be able to handle patient aggression, which may negatively influence their clinical performance and education [23].

In Saudi Arabia, the health care system comprises both the public (governmental) and private sectors; 80% of health care services fall under the public sector while 20% falls under the private sector [24]. Dental colleges under these public and private sectors are accessible to people of all nationalities. To the best of our knowledge, there were no previous studies reporting on patient aggression among dental students in Saudi Arabia. Therefore, the aim of this study is to report experience of patient aggression among dental students in Saudi Arabia and determine their reactions toward these situations and the impact these experiences have on their academic or clinical performance.

## 2. Materials and Methods

### 2.1. Study Design

A cross-sectional analytical study was conducted among dental students in Saudi Arabia. Before the start of this study, research protocol was submitted to the Institutional Review Board, King Abdullah International Medical Research Center, Riyadh, Saudi Arabia (KAIMRC). KAIMRC and the College of Dentistry, King Saud Bin Abdulaziz University for Health Sciences are under the umbrella of the Ministry of National Guard Health Affairs. The data collection process was initiated after obtaining ethical clearance (Ref. No. IRBC/1449/21).

#### 2.1.1. Sample Size and Sampling Frame

The sample size required for this study was estimated to have a power of 80% and a 95% confidence interval for a prevalence of 50% based on previous studies reported in the literature [17,25,26]. In health universities in Saudi Arabia, variation in student enrollment occurs every year, and private-institution-related data is not easily accessible; therefore, to overcome these limitations, an infinite sample was considered, following which the sample size required for this study was 355. There are 24 dental colleges in Saudi Arabia, of which 18 are governmental and 6 are private. The country was divided into 5 zones: central zone, northern zone, southern zone, western zone, and eastern zone. From each zone, universities with dental colleges were selected using a simple cluster random sampling technique. King Saud University and Qassim University represented the Center of the Kingdom, Imam Abdulrahman Bin Faisal University represented the Eastern Region, King Abdulaziz University represented the Western Region, Jazan University represented the Southern Region, and Hail University represented the Northern Region. (Figure 1).

#### 2.1.2. Inclusion and Exclusion Criteria

Inclusion criteria: Dental students in their clinical years and those willing to participate by providing a written informed consent form were included in this study.

Exclusion criteria: Dental students in their pre-clinical years and those suffering mental illnesses or psychological disorders were excluded.

#### 2.1.3. Data Collection

Study participants were informed that participation in this study was voluntary and anonymous. Data was collected using a structured Web-based Questionnaire. The questionnaire was developed and distributed in the English language after referral to similar studies reported in the literature [17,23]. The questionnaire comprised 22 questions in four main sections, each with a screening question. The first section contained demographic information (Q1–Q5), the second section contained questions related to experience with patient aggression (Q6–Q11), the third section included questions related to impact of patient aggression (Q12–Q15), and the fourth section consisted of questions related to response to patient aggression (Q16–Q22).

#### 2.1.4. Validity and Reliability of Questionnaire

Content validity of the questionnaire was assessed by a panel of experts comprising five faculty members with substantial research experience from the College of Dentistry, King Saud Bin Abdulaziz University for Health Sciences; each held an academic degree of Master’s or Doctor of Philosophy in Dental Public Health. The main purpose of this assessment was to determine whether answers to questions had a great degree of agreement among the panel of experts and to quantify concordance between panel members for each question using Aiken’s V test [27]. A value higher than 0.90 was obtained for the included questions in the questionnaire.

Reliability of the questionnaire was assessed through conduction of a pilot study of 5% of the population required for this study, in which dental students from the College of Dentistry, King Saud Bin Abdulaziz University for Health Sciences, were invited to participate. Test–retest reliability was assessed through obtained responses from the same participants before and after a gap of two weeks. An intraclass correlation coefficient value of 8, which indicated good reliability, was obtained. The pilot study responses revealed no need for further modification of the questionnaire; therefore, all 22 questions were used in the final version of the questionnaire. The responses of pilot study participants were included in the final data analysis.

#### 2.1.5. Participants

Contact details of faculty members from selected representative colleges were collected from their respective college websites; those faculty members were further requested to share students’ contact details for the purpose of distribution of invitations to participate in this study. Dental students from the selected universities were invited to participate in this study. The questionnaire was distributed online through email and on social media platforms, such as Twitter^®^, WhatsApp^®^, Telegram^®^, Instagram^®^, and Snapchat^®^ (Figure 1). The online method of data collection was adopted due to COVID-19 restrictions in the kingdom of Saudi Arabia. The data collection process was scheduled over a period of three months (from 15 September 2021 until 15 December 2021).

#### 2.1.6. Statistical Analysis

Data were analyzed using SPSS 27.0 (IBM Corporation, Armonk, NY, USA). Descriptive statistics were calculated, and chi-square analysis was used to compare the prevalence of patient aggression experiences among age groups, genders, and universities.

## 3. Results

A total of 375 participants, which was higher than the required sample, responded to the questionnaire. Demographic details of study participants are presented in (Table 1).

### 3.1. Experience of Patient Aggression among Study Participants

A total of 121 (32.3%) participants reported experiencing patient aggression. Out of these, 91 (75.21%) participants stated that they had experiences with patients displaying anger or raising their voice toward them. Moreover, 37 (30.58%) participants reported being insulted by a patient (e.g., swearing or racist words), 22 (18.18%) participants reported being threatened, 4 (2.48%) participants reported being physically harassed, 12 (6.2%) had experienced sexual harassment, and 65 (50.41%) had experienced verbal harassment. Furthermore, 91 (75.21%) participants reported being abandoned by patients because they were students, and 14 (11.57%) participants stated that they had experienced patient aggression due to their racial background, such as their last name (Table 2).

### 3.2. Impact of Patient Aggression on Study Participants

Among 121 (32.3%) participants who had experienced patient aggression, 55 (45.45%) stated that aggressive patient behavior had an impact on their clinical performance, whereas 66 (54.55%) said that it did not. Furthermore, 38 (31.4%) participants believed that their communication with other patients had been affected due to encounters with aggressive patients. Regarding psychological health, 44 (36.36%) participants considered themselves stressed out, 18 (14.88%) felt that their confidence was affected, and 14 (11.57%) felt that they were depressed. A total of 16 (13.22%) participants felt that their reputation within their institution was affected, while 105 (86.78%) participants reported that their reputations were not affected by patient aggression (Table 3).

### 3.3. Response of Study Participants towards Patients’ Aggressive Behavior

A total of 29 (23.97%) participants ignored patients with aggressive behavior, 12 (9.92%) responded aggressively, 62 (51.24%) tried to calm the patients, and 18 (14.88%) felt that they needed action to be taken. Moreover, only 27 (22.31%) participants reported any incident. A total of 18 (66.67%) participants reported patients’ aggressive behavior to their academic mentors, 6 (22.22%) reported to their respective colleges’ clinical affairs departments, and 3 (11.11%) reported to student affairs supervisors. In addition, 94 (77.69%) participants felt that there was no need to report this behavior for the following reasons: 13 (14%) participants were afraid of the negative consequences, 17 (18.3%) felt that they were not encouraged to report this behavior, 61 (65.6%) felt that the situation was not important enough to be reported, 2 (2.20%) felt ashamed or guilty, 25 (26.9%) did not know to whom to report this aggressive behavior, and 7 (7.5%) gave other answers. Moreover, 24 (19.83%) participants took time off due to incidents, and 97 (80.17%) felt that there was no need to do so. Furthermore, 22 (18.18%) participants thought about quitting dentistry due to these experiences (Table 4).

### 3.4. Association between Patient Aggression, Clinical Years, and Age Groups

The chi-square test showed that the majority of dental students in the age group of less than 20 years experienced a higher rate of mental or psychological disorders (χ^2^ = 7.41; *p* = 0.024), experienced patient aggression due to their racial background (χ^2^ Value = 7.7; *p* = 0.021), and reported that patients’ aggressive behavior affected their reputation in their institution, as compared with other age groups (χ^2^ value = 6.63; *p* = 0.036). A total of 60 (54.1%) participants aged 21–25 years experienced verbal harassment, and 5 (55.6%) and 2 (22.2%) participants older than 25 years experienced verbal and sexual harassment, respectively (χ^2^ = 36.32; *p* = 0.001). The chi-square test showed that students in their first clinical year experienced aggressive patient behavior that affected their reputation in their institution more frequently than did senior students (χ^2^ Value = 10.29, *p* = 0.006). Similarly, more students in their first clinical year thought about quitting dentistry due to patient aggression compared to senior students (χ^2^ Value = 6.16, *p* = 0.046) (Table 5).

## 4. Discussion

Experience of violence and aggression toward health care providers at workplaces is considered a global issue. Several studies have reported experience of patient aggression among nursing and medical students, since they are in direct contact with both patients and patients’ visitors and are at a higher risk of experiencing episodes of aggression and violence [28,29]. Episodes of aggression and violence towards health care providers and students may have a negative impact—including depression, anxiety, loss of self-confidence, or humiliation—on their overall wellbeing; all of these consequences can ultimately impact quality of patient care [30]. In the present study, we intended to report experience of patient aggression among dental students in Saudi Arabia, its impact on those students’ well-being and performance, and those students’ responses to these situations. In this study, only two participants were of other nationalities; this could be mainly because the majority of dental colleges in Saudi Arabia admit only students with Saudi nationality.

### 4.1. Experience of Patient Aggression among Study Participants

Experience of patient aggression has been widely reported among health-professional students. In the present study, 32.3% of study participants reported experiencing patient aggression. These findings were similar to the reported experiences of patient aggression among other health-care-professional students. Hunter EJ et al. reported experience of patient aggression among Scottish nursing students; prevalence of verbal aggression was 76.8% and physical violence was 69.6% among study participants [31]. Another similar study, conducted by Ozcan, k.N. et al., among nursing students in Istanbul reported that 64.4% of study participants had experienced some form of patient aggression [32]. Xie, Z. et al., reported experience of patient aggression among Chinese medical students, where 30.6% of study participants reported having suffered at least one patient-initiated aggression during the previous year [23]. Rhoades, K.A. et al. reported experience of patient aggression among dental students in the United States; 28% of study participants reported experiencing at least one instance of physical aggression and 86% reported experiencing at least one instance of verbal aggression [17]. A study conducted by Liaw, R.D.Y. et al. reported harassment of dental students by their patients; 20% of study participants reported being harassed. Among these, 53% had experienced verbal harassment and 19% had experienced sexual harassment. Patients sending pictures of their private body parts, making racial jokes and insults, and raising their voices were reported by these students [19]. In the present study, 6.2% of study participants reported being sexually harassed and 50.41% had experienced verbal harassment. These findings are similar to findings of the study conducted by Garbin, C.A.S. et al. among dental students in Brazil. The aforementioned authors also reported that 9.1% of study participants had experienced sexual harassment, and these students’ proximity to patients was misunderstood as intimacy [18]. Another study conducted by Zurayk, L.F. et al. among maxillofacial practitioners and residents reported that 44% of study participants were sexually harassed by their patients [33]. These higher rates of prevalence of patient aggression among health care students could be occurring due to students’ perception of having little or no power, leading to their inability to report or address these situations. This could also explain the higher prevalence among less-experienced students. These experiences may eventually have a negative impact on those students’ view of the profession [34].

### 4.2. Impact of Patient Aggression on Study Participants

Patient aggression had a negative impact on the overall well-being of health care providers [3,35,36]. Most of the time, it was reflected in the form of anxiety and depressive symptoms [37,38]. Episodes of patient aggression and violence also occasionally interfered with diagnosis and treatment planning [39,40]. In the present study, (10.10%) of students reported that their communication with other patients was affected by their previous experiences with patient aggression. Moreover, (13.22%) of students reported that their reputations were affected by patients’ aggressive behavior. These findings are similar to those of a study conducted by Zurayk, L.F. et al. among maxillofacial practitioners and residents, in which 17% reported that harassment experiences had interfered with their ability to work and 45% reported that these experiences had negatively affected their confidence [33]. Liaw, R.D.Y. et al., reported that students who had experienced patient harassment lost interest in social life and their will to continue studying dentistry [19]. These findings were similar to the results of a study conducted by Tee, S. et al., in which 81.1% of nursing students from the United Kingdom reported being psychologically and emotionally affected by patients’ aggressive behavior [41]. In the present study, 19.83% of study participants had taken time off from clinical sessions as a result of patients’ aggressive behavior. Zurayk, L.F. et al. reported that 9% of their study participants who had experienced patient harassment wished to discontinue their profession and pursue another [33]. Studies have shown that episodes of patient aggression may also lead to discontinuation of the profession and to fewer pre-professional students willing to enroll in medical professions [23,42]. Experience of patient aggression and violent behaviors may also hamper the performance of students; lead to anger, anxiety, shame, frustration, stress, flashbacks, lower commitment to work, and post-traumatic stress disorder; and result in adopting adverse habits such as tobacco abuse, alcoholism, and drug abuse [14,43,44,45,46,47].

### 4.3. Reponses of Study Participants towards Patient Aggression

Experience of violence and aggressive behavior is found among all health care professionals; however, appropriate training and polices that can help in preventing and managing these episodes are lacking [47]. In the present study, 22.31% of students reported these incidents to their academic mentors, colleges’ clinical affairs departments, or student affairs supervisors. Garbin, C.A.S. et al., reported that 26% of participants who had experienced sexual harassment from their patients ignored the incidents and only 10.5% reported the incidents to their tutors [18]. Zurayk, L.F. et al. reported that only 33% of their study participants who had experienced patient harassment reported the incident to someone in authority. A few of the reasons for not reporting the incidents included worry about the negative effect reporting may have had on the participants’ evaluation, confidentiality, and reputation; fear of being doubted; and feeling guilty and ashamed [33]. Furthermore, Cheung, K. et al. reported that only 8.25% of nursing students who experienced patient aggression reported the incident to a senior staff member [48]. Tee, S. et al. reported that nursing students never reported incidents of violence or aggression [41]. Considering these findings, we feel that there is a need to develop an agreed-upon policy in each workplace regarding how to manage incidents of patient aggression and violence.

The present study had few limitations, one of which was uneven distribution of the sample, since the data-collection form was disseminated through online portals due to COVID-19 restrictions. Another limitation was related to comparison of our results with the results of the studies conducted among other health care disciplines, such as nursing and medical students, due to lack of reported studies among dental students.

## 5. Conclusions

Prevalence of patient aggression was significant among dental students in Saudi Arabia. These findings were similar to the findings of studies reported from different regions of the world. These episodes of patient aggression have a negative impact on students’ academic performance and wellbeing, thus necessitating urgent attention. Students need to be trained in patient-centered communication skills, which can help them develop better rapport and higher trust and make them more proficient when interacting with patients. Students should be oriented regarding clinical threats and violence risks in clinical settings and must be trained to manage difficult situations and understand strategies required for managing escalation of these situations. Educational institutions should conduct periodic workshops for students and faculty in order to address these issues and should also ensure that mechanisms that facilitate reporting and dealing with these situations are in place. Policymakers should develop better policies in order to reduce violence and aggression against health-professional students.

## Figures and Tables

**Figure 1 healthcare-10-02239-f001:**
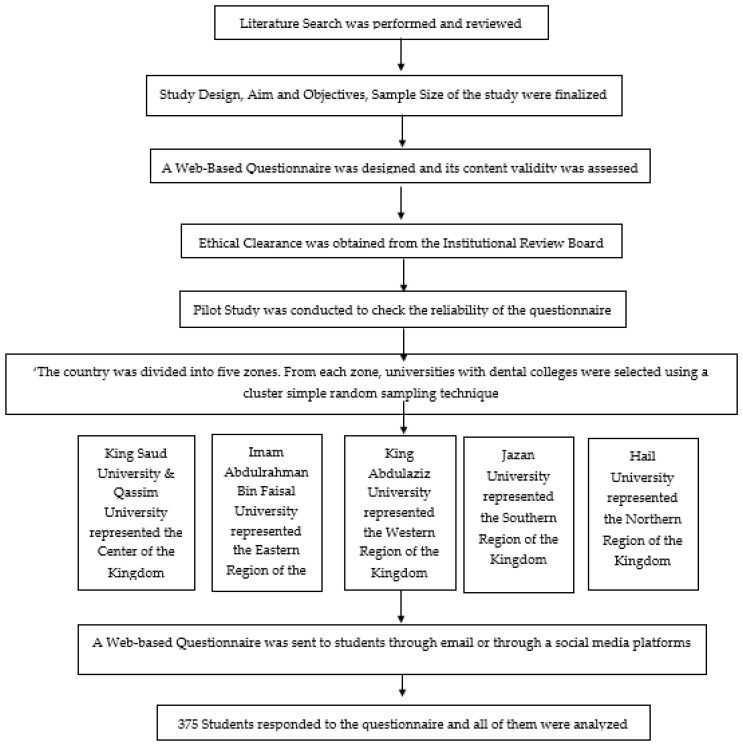
Flow chart for methodology.

**Table 1 healthcare-10-02239-t001:** Demographic details of study participants.

Parameters	Sub-Parameters	Number	Percentage
Age (years)	≤20	9	2.4%
	21–25	350	93.3%
	≥25	16	4.3%
Gender	Male	190	50.7%
	Female	185	49.3%
Geographic regions	Central	135	36%
	Western	84	22.4%
	Eastern	47	12.5%
	Northern	43	14.9%
	Southern	52	13.9%
Nationality	Saudi	373	99.5%
	Non-Saudi	2	0.5%
Clinical year	First	63	16.8%
	Second	98	26.1%
	Third	214	57.1%

**Table 2 healthcare-10-02239-t002:** Experience of patient aggression among study participants.

Question	No	Yes
Have you ever experienced patients with anger and/or raising their voice?	30 (24.79%)	91 (75.21%)
Have you ever been insulted, such as swearing and racist words, by a patient?	84 (69.42%)	37 (30.58%)
Have you ever been threatened by a patient?	99 (81.82%)	22 (18.18%)
Have you ever been intentionally abandoned by a patient because you are a student?	30 (24.79%)	91 (75.21%)
Have you ever experienced patient aggression due to your racial background, such as based on your last name?	107 (88.43%)	14 (11.57%)
Have you ever been harassed by a patient? (You can choose more than one answer.)	No	Physical harassment
	49 (40.5%)	4 (2.48%)
	Sexual harassment	Verbal harassment (criticize your treatment or clinical work)
	12 (6.2%)	65 (50.41%)

**Table 3 healthcare-10-02239-t003:** Impact of patient aggression on study participants.

Questions	No	Yes
Did the patient’s aggressive behavior have an effect on your clinical performance?	66 (54.55%)	55 (45.45%)
Did the patient’s aggressive behavior have an effect on your communication with other patients?	83 (68.6%)	38 (31.4%)
Did the patient’s aggressive behavior affect your reputation in your institution?	105 (86.78%)	16 (13.22%)
Did the patient’s aggressive behavior affect your psychological health?	Made me stressed	Affected my confidence
	44 (36.36%)	18 (14.88%)
	Made me depressed	It did not affect me
	16 (13.22%)	42 (34.71%)
	Other	
	3 (2.49%)	

**Table 4 healthcare-10-02239-t004:** Response of study participants towards patients’ aggressive behavior.

Questions	No	Yes
Have you ever taken time off from the clinical sessions as a result of patient’s aggressive behavior?	97 (80.17%)	24 (19.83%)
Have you ever thought about quitting dentistry as a result of patient aggression?	99 (81.82%)	22 (18.18%)
Have you tried to report the patient’s aggressive behavior?	94 (77.69%)	27 (22.31%)
If yes, then who did you report it to?	Academic mentor	Patient clinical affairs
	18 (66.67%)	6 (22.22%)
	Student affairs supervisor	
	3 (11.11%)	
If no, what was the reason?	I was afraid of the negative consequences	Nobody encouraged me
	13 (14%)	17 (18.3%)
	It was not important	Felt ashamed or guilty
	61 (65.6%)	2 (2.20%)
	I Did not know who to report to	Others
	25 (26.9%)	7 (7.5%)
What was your response to the patient’s aggressive behavior?	I ignored the patient	I responded aggressively
	29 (23.97%)	12 (9.92)
	I tried to calm the patient	No action was taken
	62 (51.24%)	18 (14.88%)

**Table 5 healthcare-10-02239-t005:** Association between patient aggression and clinical years and age groups.

		Age Groups						χ^2^ Value	*p* Value
		Younger than 20 Years		21–25 Years		Older than 25 Years			
		N	%	N	%	N	%		
Have you ever been diagnosed with any mental or psychological disorder?	No	0	0.00%	98	88.30%	7	77.80%	7.419	0.024 *
	Yes	1	100.00%	13	11.70%	2	22.20%		
Have you ever experienced patient aggression because of racial background?	No	0	0.00%	99	89.20%	8	88.90%	7.707	0.021 *
	Yes	1	100.00%	12	10.80%	1	11.10%		
Did the patient’s aggressive behavior affect your reputation in your institution?	No	0	0.00%	97	87.40%	8	88.90%	6.634	0.036 *
	Yes	1	100.00%	14	12.60%	1	11.10%		
Have you ever been harassed by the patient? (You can choose more than one answer.)	Psychological harassment (criticize your treatment and clinical work)	0	0.00%	60	54.10%	5	55.60%	36.324	0.001 *
	Physical harassment	1	100.00%	2	1.80%	1	11.10%		
	Sexual harassment	0	0.00%	10	9.00%	2	22.20%		
	I have never been harassed by a patient	1	100.00%	46	41.40%	3	33.30%		
		What is your current academic year?						χ^2^ Value	*p* Value
		First clinical year		Second clinical year		Third clinical year			
		N	%	N	%	N	%		
Did the patient’s aggressive behavior affect your reputation in your institution?	No	4	50.00%	23	92.00%	78	88.60%	10.289	0.006 *
	Yes	4	50.00%	2	8.00%	10	11.40%		
Have you ever thought about quitting dentistry due to patient aggression?	No	4	50.00%	22	88.00%	73	83.00%	6.163	0.046 *
	Yes	4	50.00%	3	12.00%	15	17.00%		

* Statistical significance set at <0.05; N: number of samples; χ^2^ value: chi-square value.

## Data Availability

Not applicable.

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
