# Peer review of "Dental Students’ Experience, Impact, and Response to Patient Aggression in Saudi Arabia: A Nationwide Study"

_healthcare, 2022, doi:10.3390/healthcare10112239_

Round 1

Reviewer 1 Report

Introduction

The introduction is too short and lack substance. What are the impication of abuse against health workers? You cited several literature to prove that healthcare workers experience violence and abuse. But the major question is ''and so what''? does this abuse have any meaning? wh cares? do we have similar study in another country of this nature?

Methods

Your decision of sample size of 355 is clumsy. Please explain why and how this was gotten. Also, confirm if this work is part of a larger work. Mere citing literature is not substantial for sample size. 

Data collection

Line 93 Data was not self administered due to COVID but rather it was administered via online (This was your claim on line 83). Please align. Also, line 94/95 do these similar studies that inform your questionnaire share the same context and demographics?

validity

Please state the qualification of those that reviewed the questionnaire. Line 103 "our institution" meaning what? are you writing for the general public or for your institution?

Reliability

Was the pilot study conducted among 5% of those who are also participants pf the main study? This is not scholarly. Kindly conduct the pilot study among other healthcare practitioners who are not part of the major study. If the questionnaire was not modified by those expert who reviewed it, was the questionnaire adopted from previous study? This is very clumsy

Participants

How many universities do we have in the country where the study was conducted? How many region do we have and how many universities do we have in each region? How was the sample unit selected? How did you get the phone number and handles of these health workers to be able to reach them via WhatsApps, Twitter,, Telegram etc. Where are the sample units? Any gender bias? any age specification for data collection?

The design for the study is clumsy and need to be revisit for global consumption

Author Response

Dear Reviewer,
Greetings!

Firstly I would like to thank you for your valuable inputs, and would like to inform you that we have considered all the valuable comments suggested by you and have modified the manuscript as per your suggestions.

We are also providing point to point clarifications for the comments suggested by you.  "Please see the attachment."

We have modified the manuscript to best of our knowledge, kindly consider the same and oblige,

Thank you and regards

Sincerely yours,

Dr. Sanjeev B Khanagar (Corresponding Author)

Reviewer 1

  1. The introduction is too short and lack substance. What are the impication of abuse against health workers? You cited several literature to prove that healthcare workers experience violence and abuse. But the major question is ''and so what''? does this abuse have any meaning? wh cares? do we have similar study in another country of this nature?

Response: We have considered the valuable suggestions, and we have added more detailed information on the experiences and impact of patient aggression and harassment on health care providers in the introduction and discussion section

  1. Your decision of sample size of 355 is clumsy. Please explain why and how this was gotten. Also, confirm if this work is part of a larger work. Mere citing literature is not substantial for sample size. 

Response: As per the statistician and based on the literature evidence, proportion was considered to be 50%. And confidence interval of 95% and infinite population sample. The calculation was done using open epi software. 

  1. Line 93 Data was not self administered due to COVID but rather it was administered via online (This was your claim on line 83). Please align. Also, line 94/95 do these similar studies that inform your questionnaire share the same context and demographics?

Response: Self-administered survey is a data-collection process where the researcher is entirely absent when respondents are filling out the survey. Self-administered also includes online method of data collection, so we have stated it as self-administered.  

Yes, the questionnaire was designed after referring the studies that were conducted among medical and dental students.

  1. Please state the qualification of those that reviewed the questionnaire. Line 103 "our institution" meaning what? are you writing for the general public or for your institution?

Response: We have added the qualifications of the reviewers (faculty members) as per reviewer’s suggestion.

  1. Was the pilot study conducted among 5% of those who are also participants pf the main study? This is not scholarly. Kindly conduct the pilot study among other healthcare practitioners who are not part of the major study. If the questionnaire was not modified by those expert who reviewed it, was the questionnaire adopted from previous study? This is very clumsy

Response: We appreciate the reviewer’s concern, however considering the time the study was conducted (during COVID-19 restrictions) reaching the study participants was difficult. So we had consulted an experienced biostatistician, and decided to conduct an internal pilot study and include the data in main, based on the statistician’s advice with reference.    https://www.ncbi.nlm.nih.gov/pmc/articles/PMC5716817/

Introduction of a pilot study. Korean J Anesthesiol. 2017 Dec;70(6):601-605. doi: 10.4097/kjae.2017.70.6.601. Epub 2017 Nov 14. PMID: 29225742; PMCID: PMC5716817.

  1. How many universities do we have in the country where the study was conducted? How many region do we have and how many universities do we have in each region? How was the sample unit selected? How did you get the phone number and handles of these health workers to be able to reach them via WhatsApps, Twitter,, Telegram etc. Where are the sample units? Any gender bias? any age specification for data collection?

The design for the study is clumsy and need to be revisit for global consumption

Response: We thank the reviewer for the valuable suggestion, we have added all the detailed information suggested by the reviewer.

Reviewer 2 Report

Thank you for an interesting read.

I think that one should describe the context better in this article, which kind of health care system (or systems, private and public?) does Saudi Arabia have and how is dentistry located in that? Who has access to the health care system, are public and private sectors different having different types of patients etc.? Which kinds of institutions form the health care system, is King Abdullah International Medical Research Center the umbrella organization for scientific study when it comes to health care or why is it involved?

Why you had chosen the specific field as your field of study, is it related to your personal professions or experiences?

Careful description of methods is missing. How exactly did you have an access to data and how did you distribute the questionnaires?

Were there any challenges when the data was gathered? What was easy and why? How did these facts affect the material gathered?

Abstract: I am not quite sure if all the numbers should be cited in abstract

Why there are no other nationalities included in the sample? Was the study conducted in Arabic or in English or in some other language? Why was this language chosen? Did you translate the material when you wrote the article?

“The pilot study responses revealed no need for further modification of the questionnaire” in which way this manifested?

“Sixteen (13.22%) participants felt that their reputation was affected by their institution”, what does this mean exactly?

I think that the names of the universities and regions where the samples were gathered should be mentioned in the text, not only in the figure, as they are important part of the context of the study. Also, one should tell if in certain area there was more data gathered in some other less, and maybe ponder on why was this

Section 4.1: I think that earlier research should be discussed earlier in the text

All in all in the analytical part I would encourage stating things based on the material in question, one can compare it to other studies, but it is important to concentrate on the fact one speaks based on one’s own material

In conclusions one could also discuss, if the results in Saudi Arabia are different than in other countries, why it is so

Author Response

Dear Reviewer,
Greetings of the day!

Firstly I would like to thank you for your valuable inputs, and would like to inform you that we have considered your valuable comments and have modified the manuscript as per your suggestions.

We are also providing point to point clarifications for the comments suggested by you. "Please see the attachment."

We have modified the manuscript to best of our knowledge, kindly consider the same and oblige,

Thank you and regards

Sincerely yours,

Dr. Sanjeev B Khanagar (Corresponding Author)

Reviewer 2

  1. I think that one should describe the context better in this article, which kind of health care system (or systems, private and public?) does Saudi Arabia have and how is dentistry located in that? Who has access to the health care system, are public and private sectors different having different types of patients etc.? Which kinds of institutions form the health care system, is King Abdullah International Medical Research Center the umbrella organization for scientific study when it comes to health care or why is it involved?

Response: We have considered the valuable suggestions, and we have added all the information suggested by the reviewer.

  1. Why you had chosen the specific field as your field of study, is it related to your personal professions or experiences?

Response: We have chosen this field because it is related to our profession and also there is no study conducted in this region on patient aggression and harassment on dental students.

  1. Careful description of methods is missing. How exactly did you have an access to data and how did you distribute the questionnaires?

Response: We have added a detailed information of the methodology after considering the valuable comments of the reviewers.

  1. Were there any challenges when the data was gathered? What was easy and why? How did these facts affect the material gathered?

Response: Thank you for your valuable suggestion, we have added these statements as the limitations of the study

  1. Abstract: I am not quite sure if all the numbers should be cited in abstract

Response: Thank you for your concern, however we had referred few previously published articles from this journal and considered mentioning the numbers in this section

  1. Why there are no other nationalities included in the sample? Was the study conducted in Arabic or in English or in some other language? Why was this language chosen? Did you translate the material when you wrote the article?

Response: We have included other nationalities in this study, and in the present study there were only 2 participants from other nationalities, this is mainly because most of the universities in Saudi Arabia admit only students with Saudi Nationality.

The study was conducted in English Language and we have added the same information in the manuscript after considering the valuable suggestion of the reviewer.

  1. “The pilot study responses revealed no need for further modification of the questionnaire” in which way this manifested?

Response: Here we wanted to know if there are any question unanswered by the participants in the pilot study. So after obtaining a good response in the pilot study we retained the same questionnaire for the main study without any further modifications

  1. “Sixteen (13.22%) participants felt that their reputation was affected by their institution”, what does this mean exactly?

Response: Thank you for highlighting this, it was a typo error and we have corrected this statement in the manuscript

Reviewer 3 Report

Dental Students’ Experience, Impact, and Response to Patient Aggression in Saudi Arabia: A Nationwide Study is a useful, important addition to the literature. There are several issues that should be addressed to increase its utility.

1.     I was surprised that the (small) literature about aggression toward licensed dentists was not mentioned. Here are the papers that have studied that:

a.     Ayers KMS, Thomson WM, Newton JT, Morgaine KC, Rich AM. Self-reported occupational health of general dental practitioners. Occup Med (Lond). 2009;59(93):142-148.

b.     Azodo CC, Ezeja EB, Ehikhamenor EE. Occupa- tional violence against dental professionals in southern Nigeria. Afr Health Sci. 2011;11(3):486-492.

c.     British Dental Association. Survey into violence and abuse against general dental practitioners and their staff. London: British Dental Association; 1997.

d.     Pemberton MN, Atherton GJ, Thornhill MH. Violence and aggression at work. Br Dent J. 2000;189:409-410.

e.     Rhoades KA, Heyman RE, Eddy JM, Haydt NC, Glazman JE, Dispirito ZF, Fat SJ, Guerrera CM, Rascon AN, Wolff MS. Patient aggression toward dentists. The Journal of the American Dental Association. 2020 Oct 1;151(10):764-9.

2.     Likewise, there was a very recently published article on harassment of dental students that the authors should be aware of: Liaw RD, Ling DC, Vuli LJ, Loch C, Adam LA. “It's just inappropriate”: Harassment of dental students by patients. Journal of Dental Education. 2022 May;86(5):605-14.     

3.     The paper was clearly written. However, there were problems with English (e.g., subject-verb agreement, usage, redundancies, punctuation) that should be corrected for maximal impact. Hiring a copyeditor would be a sensible option.

4.     “Psychosocial harassment” is mentioned but never defined.

5.     “Psychological harassment (criticize your treatment or clinical work)” is poorly operarationalized. How do we know this is true harassment and not complaints about badly completed work from students who are learning?

6.     The authors use “reliability” in a way that does not fit traditional psychometrics definitions of test-retest reliability, internal consistency, or other ways of measuring stability of measurement. I do not understand what they meant by the following: “Reliability of the questionnaire was assessed by conducting a pilot study among 5% of 108 the study population required for this study, for which dental students from our institution were invited to participate. The pilot study responses revealed no need for further modification of the questionnaire; therefore, all 22 questions were used in the final version 111 of the questionnaire. The responses of the pilot study participants were included in the 112 final data analysis.” Did they mean that 5% took the questionnaire and understood it? Please clarify (and likely remove the word “reliability”).

7.     Why is “abandonment” in this paper? Are we to believe that patients who do not return — because they are dissatisfied from their experience at a dental school, have dental fear, or a million other reasons — are considered to commit aggression for their termination of the professional relationship?

Author Response

Dear Reviewer,
Greetings!

Firstly I would like to thank you for your valuable inputs, and would like to inform you that we have considered all the valuable comments suggested by you and have modified the manuscript as per your suggestions.

We are also providing point to point clarifications for the comments suggested by you.  "Please see the attachment."

We have modified the manuscript to best of our knowledge, kindly consider the same and oblige,

Thank you and regards

Sincerely yours,

Dr. Sanjeev B Khanagar (Corresponding Author)

Reviewer 3

  1. I was surprised that the (small) literature about aggression toward licensed dentists was not mentioned. Here are the papers that have studied that:
  2. Ayers KMS, Thomson WM, Newton JT, Morgaine KC, Rich AM. Self-reported occupational health of general dental practitioners. Occup Med (Lond). 2009;59(93):142-148.
  3. Azodo CC, Ezeja EB, Ehikhamenor EE. Occupa- tional violence against dental professionals in southern Nigeria. Afr Health Sci. 2011;11(3):486-492.
  4. British Dental Association. Survey into violence and abuse against general dental practitioners and their staff. London: British Dental Association; 1997.
  5. Pemberton MN, Atherton GJ, Thornhill MH. Violence and aggression at work. Br Dent J. 2000;189:409-410.
  6. Rhoades KA, Heyman RE, Eddy JM, Haydt NC, Glazman JE, Dispirito ZF, Fat SJ, Guerrera CM, Rascon AN, Wolff MS. Patient aggression toward dentists. The Journal of the American Dental Association. 2020 Oct 1;151(10):764-9.

Response: We sincerely thank the reviewer for the valuable suggestions and recommending us the valuable data sources.

We have included all the articles suggested by the reviewer and also a added few additional articles after screening the reference list of these articles.

We have revised the major sections of our manuscript with this new information provided by the reviewer.

  1. Likewise, there was a very recently published article on harassment of dental students that the authors should be aware of: Liaw RD, Ling DC, Vuli LJ, Loch C, Adam LA. “It's just inappropriate”: Harassment of dental students by patients. Journal of Dental Education. 2022 May;86(5):605-14.

Response: We have included all the articles suggested by the reviewer     

  1. The paper was clearly written. However, there were problems with English (e.g., subject-verb agreement, usage, redundancies, punctuation) that should be corrected for maximal impact. Hiring a copyeditor would be a sensible option.

Response: We thank you for the words of appreciation, the revised manuscript is reviewed and edited by an Expert who is proficient in English Language. 

  1. & 5 “Psychosocial harassment” is mentioned but never defined. “Psychological harassment (criticize your treatment or clinical work)” is poorly operarationalized. How do we know this is true harassment and not complaints about badly completed work from students who are learning?

Response: We thank you for the valuable inputs, “Psychosocial harassment” (criticize your treatment or clinical work)” was actually a form of assessing verbal harassment. So we have changed it to Verbal Harassment in order to be more clear.

  1. The authors use “reliability” in a way that does not fit traditional psychometrics definitions of test-retest reliability, internal consistency, or other ways of measuring stability of measurement. I do not understand what they meant by the following: “Reliability of the questionnaire was assessed by conducting a pilot study among 5% of 108 the study population required for this study, for which dental students from our institution were invited to participate. The pilot study responses revealed no need for further modification of the questionnaire; therefore, all 22 questions were used in the final version 111 of the questionnaire. The responses of the pilot study participants were included in the 112 final data analysis.” Did they mean that 5% took the questionnaire and understood it? Please clarify (and likely remove the word “reliability”).

Response: We thank you for the valuable inputs, we have revised the section for better clarity

  1. Why is “abandonment” in this paper? Are we to believe that patients who do not return — because they are dissatisfied from their experience at a dental school, have dental fear, or a million other reasons — are considered to commit aggression for their termination of the professional relationship?

Response:  In this paper “Abandonment” meant- a student “Being Rejected” by the patient after knowing that he is still an undergraduate student.

Round 2

Reviewer 1 Report

The use of Infinite Population to describe Dental Students is debatable. It is advisable to search the literature extensively for recent usage before application.  However, the recent version is an improvement. 

Author Response

Dear Reviewer,
Greetings!

Firstly I would like to thank you for your valuable inputs, and would like to inform you that we have considered all the valuable comments suggested by you and have modified the manuscript as per your suggestions.

We are also providing point to point clarifications for the comments suggested by you.  "Please see the attachment."

We have modified the manuscript to best of our knowledge, kindly consider the same and oblige,

Thank you and regards

Sincerely yours,

Dr. Sanjeev B Khanagar (Corresponding Author)

Reviewer 1

The use of Infinite Population to describe Dental Students is debatable. It is advisable to search the literature extensively for recent usage before application.  However, the recent version is an improvement. 

Response: We understand and thank you for your valuable suggestion, however “In the health universities in Saudi Arabia, variation in student enrollment occurs every year and private institution related data is not easily accessible, hence to overcome these limitations an infinite sample was considered, following which the sample size required for this study was 355.

We have considered your valuable suggestion and added the same in the manuscript (2.1.1 Sample size and sampling frame)

Reviewer 2 Report

I have read the revised text, and the authors' response, some of the revisions that I suggested regarding the context of the study and the methodology are not visible in the text even though they are discussed in the response such as more careful description of the context of the study, the language used in the questionnaire and the fact that why only Saudi citizens are part of the study and how did one exclude from the study certain groups of students. However, I am a scholar of cultural studies, so I might be interested in issues that are not of interest for scholars of other fields and might not be of interest for the publications which focus on health.

Author Response

Dear Reviewer,
Greetings of the day!

Firstly I would like to thank you for your valuable inputs, and would like to inform you that we have considered your valuable comments and have modified the manuscript as per your suggestions.

We are also providing point to point clarifications for the comments suggested by you. "Please see the attachment."

We have modified the manuscript to best of our knowledge, kindly consider the same and oblige,

Thank you and regards

Sincerely yours,

Dr. Sanjeev B Khanagar (Corresponding Author)

Reviewer 2

I have read the revised text, and the authors' response, some of the revisions that I suggested regarding the context of the study and the methodology are not visible in the text even though they are discussed in the response such as more careful description of the context of the study, the language used in the questionnaire and the fact that why only Saudi citizens are part of the study and how did one exclude from the study certain groups of students. However, I am a scholar of cultural studies, so I might be interested in issues that are not of interest for scholars of other fields and might not be of interest for the publications which focus on health.

Response: We sincerely thank you for your valuable suggestions. We have considered the valuable suggestions, and we have added the information in the manuscript (2.1.3. Data collection) that the questionnaire was developed and distributed in English language.

We have also added a statement in the discussion section that “In the present study we intended to report on the experience of patient aggression among dental students in Saudi Arabia, its impact on the students’ well-being and performance, and their response to these situations. However, in this study only two participants were from other nationalities, this could be mainly because, majority of the dental colleges in Saudi Arabia admit students only with Saudi nationality.”  
